# The Safety of Cold-Chain Food in Post-COVID-19 Pandemic: Precaution and Quarantine

**DOI:** 10.3390/foods11111540

**Published:** 2022-05-24

**Authors:** Jia Kong, Wenxin Li, Jinyao Hu, Shixuan Zhao, Tianli Yue, Zhonghong Li, Yinqiang Xia

**Affiliations:** 1College of Food Science and Engineering, Northwest A&F University, Xianyang 712100, China; kongjia@nwafu.edu.cn (J.K.); 2019013177@nwafu.edu.cn (W.L.); hujinyao@nwafu.edu.cn (J.H.); zhaoshixuan@nwafu.edu.cn (S.Z.); yuetl305@nwsuaf.edu.cn (T.Y.); steveli@nwsuaf.edu.cn (Z.L.); 2Laboratory of Quality & Safety Risk Assessment for Agro-Products, Ministry of Agriculture, Xianyang 712100, China

**Keywords:** food safety, SARS-CoV-2, precaution, quarantine, cold-chain foods

## Abstract

Since the outbreak of coronavirus disease-19 (COVID-19), cold-chain food contamination caused by the pathogenic severe acute respiratory syndrome coronavirus-2 (SARS-CoV-2) has attracted huge concern. Cold-chain foods provide a congenial environment for SARS-CoV-2 survival, which presents a potential risk for public health. Strengthening the SARS-CoV-2 supervision of cold-chain foods has become the top priority in many countries. Methodologically, the potential safety risks and precaution measures of SARS-CoV-2 contamination on cold-chain food are analyzed. To ensure the safety of cold-chain foods, the advances in SARS-CoV-2 detection strategies are summarized based on technical principles and target biomarkers. In particular, the techniques suitable for SARS-CoV-2 detection in a cold-chain environment are discussed. Although many quarantine techniques are available, the field-based quarantine technique on cold-chain food with characteristics of real-time, sensitive, specific, portable, and large-scale application is urgently needed.

## 1. Introduction

To date, more than five hundred million people have been confirmed cases of COVID-19, which has caused over six million deaths [1]. COVID-19 caused by SARS-CoV-2 has led to a severe threat to public health and safety. At present, SARS-CoV-2 is mainly transmitted via respiratory aerosols, droplets, and close contact with SARS-CoV-2-infected patients, which can be effectively prevented through protective measures [2,3]. Unfortunately, many challenging problems in dealing with the spread of COVID-19 are emerging, e.g., a crop of the more transmissible form of SARS-CoV-2 variants, spreading of the asymptomatic carrier, and exposure to latent polluted objects. As the levels of willingness to accept the SARS-CoV-2 vaccine are insufficient to achieve community immunity and no specific drug treatment is available for the epidemic till now [4,5], many countries have adopted a policy of closure and continuous testing of a potentially infected person and contaminable objects [6].

Cold-chain foods can act as the potential carrier of COVID-19. Workers with the COVID-19 symptom may contaminate the food that is being processed [7]. Even worse, SARS-CoV-2 remains highly stable on fish, meat, poultry, pigskin, and other foods under cold storage (4 °C) and in freezing conditions (−80 °C) that can survive on cold-chain food for more than 21 days. This caused a huge risk of long-distance transmission through contaminated cold-chain foods [8]. Although it is unclear whether the viral load on the cold-chain foods is sufficient to cause an infection, the transmission risk caused by contaminated foods to humans exists [9]. Among the cases of SARS-CoV-2 contamination on cold-chain food, one of the most representative outbreaks was in Qingdao, China, in September 2020 [10,11]. After testing positive for nucleic acid, the stevedores at Qingdao Port were diagnosed with asymptomatic SARS-CoV-2 infection. Both patients began unloading frozen cod in bulk on 19 September 2020. However, neither two cases have a contact history with COVID-19 or lived abroad. Notably, the virus identified in the stevedores was highly homologous to the disease virus on the packaging of the frozen cod [12]. This finding suggests that it is possible to spread the SARS-CoV-2 through contact with contaminated cold-chain foods. Moreover, SARS-CoV-2 has been successively isolated from cold-chain food in nine provinces of China [12,13]. Countries at a low infected level or stable epidemic situation should pay more attention to the risk of imported cases to avoid another major outbreak. Therefore, it is urgent to understand the characteristics of SARS-CoV-2 transmission via cold-chain routes [14]. Strengthening the inspection and quarantine of cold-chain foods should be one of the top priorities for COVID-19 prevention.

## 2. Safety Precautions in Cold-Chain Links

A complete cold-chain is an uninterrupted process of cold production, storage, and distribution activities, along with associated equipment and logistics to keep a low ideal temperature range to ensure the quality of the transported goods [15]. However, the low temperature provides an ideal opportunity for long-distance transmission of SARS-CoV-2. Cold-chain food samples that contain a high viral concentration or prolonged exposure to contaminated food cause a risk of SARS-CoV-2 transmission from food products to persons. As such, the potential safety risks and precaution measures during the cold-chain links including the acquisition of raw materials, the processing, and treatment of raw materials, the packaging of processed food, the transportation of processed food, the sales of products, and the final consumers’ preservation should be sequentially analyzed (Figure 1) [16].

### 2.1. Acquisition of Food Raw Materials

SARS-CoV-2 contamination may originate from the acquisition of raw materials of agricultural and sideline products. Most of the people involved in this link are grass-roots producers, who have insufficient safeguard procedures and understanding of the huge risks brought by SARS-CoV-2. Once the producer is infected, they cannot diagnose themselves during the latent period and insist on productive activities as usual. Indeed, a more obvious danger is that the producers do not seek proper hospital treatment in time after symptoms of COVID-19. During this period, the sick producers may contaminate the food raw materials easily [17]. Several reasons, such as poor financial situation and individual consciousness, cause this dilemma. In addition, a lack of producer materials may also cause the transmission risk in the production activity. For example, using untreated excreta of pit latrines as fertilizer is a common phenomenon in rural agriculture, which may cause a sustainable source of infection when dealing with COVID-19 [18].

### 2.2. Processing of Food Raw Materials

For the workers in the industry, observing the personal hygiene principles, such as being equipped with strict protective measures, and identification of suspicion in production lines, are important measures since a large number of raw materials may contain contaminated agricultural production and then cause transmission risk by improper processing activities. Food sanitizing with light (e.g., ultraviolet), surface sanitizing with chemical disinfector (e.g., medicinal alcohol), and providing a harsh environment for SARS-CoV-2 survival (e.g., heat treatment) in food production lines play a positive role in preventing diseases [19]. For food enterprises, it is beneficial to strengthen the implementation of enterprise health and management during the COVID-19 pandemic. It is crucial to prevent cross-infection among infected workers, contaminated food, and contaminated machines as aerosols carrying the virus can affect workers in poorly ventilated and dense environments. Several food operations technologies, such as avoidance of acquisition of any raw food that has a potential source of SARS-CoV-2 and encouragement of food production that enriches with vitamin D, C, B3, K, amino acid L-tryptophan, and nicotinamide adenine dinucleotide (NAD+), should be advocated [20].

### 2.3. Food Packaging

In this link, packaging materials have an important role in disease prevention. SARS-CoV-2 and other CoVs have remarkably short persistence on low porosity materials (e.g., copper and latex) as compared to high porous fabrics surfaces like stainless steel, plastics, and glass. In general, SARS-CoV-2 can survive for 3–4 days on a plastic surface and retain its infectivity [21]. Viral particles decayed after 7 days on the plastic surface at room temperature at 65% relative humidity (RH), and the viability of the SARS coronavirus was significantly affected by temperature and RH [22,23]. In another study, SARS-CoV-2 persisted for shorter periods on copper, copper-nickel, and brass than that on plastic (e.g., coronaviruses survive on copper for 8 h) [24]. Therefore, the selection of appropriate materials for cold-chain food packaging can effectively reduce the probability of virus infection. In addition, food packaging sanitizing is of great importance in this cold-chain link.

### 2.4. Food Transportation

Food transportation is one of the largest cross-regional links in cold-chain procedures. Once the food contamination comes into being, various species of viruses from different regions flow everywhere, which easily causes infection and even an outbreak of epidemic. In this link, strengthening the SARS-CoV-2 test on cold-chain foods is one of the most effective measures [25]. Thus, choosing the appropriate detection strategies of SARS-CoV-2 based on technical principles can effectively increase diagnosis rates and control the risk of infection. For this purpose, the strategy of SARS-CoV-2 detection on cold-chain food with characteristics, e.g., real-time, sensitive, specific, and large-scale application, is urgently needed [26,27]. Continuous testing for cold-chain food samples, especially foods that come from countries with high counts of infected cases, is an important technological approach that ensures food security. In addition, banning the cold-chain food imports after packaging tests positive for COVID-19 is a way to cut off the transmission.

### 2.5. Sales of Food

For food sales, this link involves the widest range of people. Whether salespeople or consumers, anyone who comes into contact with contaminated cold-chain food has a probability of becoming a patient through close contact. The Brazilian Association of Supermarkets launched a pamphlet with strategies to decrease the risk of infection, emphasizing the need for sanitization of parking meters, supermarket carts, and baskets, and points of intense and repetitive contact such as door handles, doorknobs, and handrails, payment terminals, and elevators [28]. The pamphlet also emphasizes the need for public dispensers of hand sanitizer at the entrance to food markets, and soap and paper towels in the restrooms. These measures are worthwhile acting as a reference in severely affected countries or regions. At the same time, the impact of the COVID-19 pandemic on food sales cannot be ignored. To minimize the sharp drop in food sales, relevant departments should provide partial economic support [29].

### 2.6. Consumption

After people consume in public places, it is not clear whether the product is contaminated or not. No evidence showed that the SARS-CoV-2 infected consumers via food or food packaging. However, taking good care of personal protection is necessary, e.g., both the consumers and the cold-chain food should be effectively disinfected. The number of SARS-CoV-2 decreased dramatically by about seven logs after heat treatment of 70 °C for 5 min [30]. Here, the commonly used food cooking temperature and the methods are as follows: for beef, pork, veal, and lamb, cook at at least 160 °F; for turkey and chicken, cook at 165 °F; steaks, roasts, and chops, cook at at least 145 °F for 3 min; all poultry (breasts, whole bird, legs, thighs, wings, ground poultry, giblets, and stuffing) should be cooked at 165 °F; fresh pork, including fresh and ham, usually needs to be cooked at 145 °F; eggs should be cooked until yolk and white are firm; clams, oysters, and mussels should be cooked until their shells open during cooking [31].

## 3. Characteristics for the Cold-Chain Food Quarantine

Detection of SARS-CoV-2 on cold-chain foods plays an important role in the identification of pathogens at an early stage [32]. For the detection of SARS-CoV-2 contamination on cold-chain foods, different kinds of methods are available, as mentioned in Section 4 [33,34]. However, these methods are mostly used for quarantine in patients, and many methods cannot be directly applied for the SARS-CoV-2 tests on cold-chain food (Figure 2). Here, to analyze what would the ideal quarantine method look like, the characteristics of the SARS-CoV-2 test on cold-chain food are discussed.

Before the detection of SARS-CoV-2 on cold-chain foods, the sample pretreatment is an essential part and is different in nucleic acid assays and immunoassays. Pesticides and other chemicals may leave residues in chilled fresh fruits and vegetables, and animal meat such as frozen fish may be parasitized by other pathogenic microorganisms [35]. These organic reagents and pathogenic microorganisms, including some viruses and bacteria, may affect the enzyme activity and the accuracy of test results. For sample pretreatment, ultra-filtration and extraction is the most frequently used method in water samples in nucleic acid assays. Elution and flocculation techniques can be applied to obtain the virus from food matrices in the pretreatment of fruits, vegetables, and their products [36]. Elution and concentration method were also used for the treatment of different meat products [37]. Additional details on sample pretreatment can be referred to the comprehensive review of ref. [38].

To avoid affecting the sale of food, it is better to ensure the integrity of the food being tested [39]. Thus, tap water that melted in ice for cold-chain transportation, soaking liquid from packaging bags of frozen food, and extracts from the surface of fresh seafood can be selected as the media related to the detection of cold-chain food. In addition, swab specimens on the surface of the food are also feasible for the detection of SARS-CoV-2, which will not affect the integrity of the food [40]. However, the above sampling method is not feasible for animal products infected with the virus before death. Besides, the virus is unevenly distributed on imported cold-chain foods and packaging, and the virus concentration usually stays at a trace level [41]. Local sampling may lead to false-negative results. A pooled testing strategy for identifying SARS-CoV-2 can be used as it can handle a large number of food samples. Pooled testing benefits from faster results, lower costs of testing, and fewer test kits which can greatly reduce test costs [42]. Therefore, it is particularly important to adopt a comprehensive sampling and evaluation method.

Whether chilled or frozen, cold-chain foods are in a low-temperature environment. Therefore, the testing temperature is also under consideration. The most ideal state is to directly perform real-time detection on-site at low temperatures. In this way, the damage to cold-chain food is minimal and the quarantine speed of food can be accelerated. However, the current quarantine techniques require the presence of enzyme reagents (e.g., polymerases, Cas12a) [43]. The working temperature of the enzyme reagent is contradictory to the cold-chain conditions. Besides, enzyme-dependent nucleic acid detection involves harsh storage and reaction conditions of the enzyme. The enzyme-dependent inspection cycle is usually long, which forms an obvious contradiction with the short shelf life of imported cold-chain food [44]. It is necessary to contemplate whether methods using enzyme reagents are suitable for on-site screening of cold-chain foods. Thus, the development of a microthermal, non-enzyme, and spontaneous detection reagent may be the future direction of the SARS-CoV-2 test on cold-chain foods [45].

At last, the epidemic prevention in the normalization stage should be to maximize rapid detection and avoid the delay in the delivery time of cold-chain food. Thus, it is better to make a field-based test for SARS-CoV-2 diagnosis on cold-chain food at the customs. In this way, the detection method that required large-scale equipment cannot work on the spot [13]. A point-of-care test (POCT) could be one way to solve this problem because the detection time is shortened and the instrument is portable. For instance, microfluidic chips that integrate various small-scale laboratory functions on a single chip may be the most suitable technique for the rapid detection of SARS-CoV-2 on cold-chain food among the SARS-CoV-2 quarantine technologies, which shows promise of being commercialized for rapid quarantine of SARS-CoV-2 in the cold-chain environment [46,47].

## 4. Potential Cold-Chain Food Quarantine Techniques

Effective control of infectious diseases relies on individual protection as mentioned in Section 2, as significant as reasonable quarantine methods. Unlike traditional food testing methods, the detection of SARS-CoV-2 on cold-chain foods puts forward higher requirements regarding sampling method, detecting conditions, testing periodic time, anti-interference capability, and portability of equipment [48]. Here, according to the different pathogen markers, e.g., nucleic acid, antigen, antibody, and cytokine storm assay, we list two categories of COVID-19 testing methods, as well as the advance of SARS-CoV-2 test methods (Table 1). Whether the existing detection methods can be applied to test suspicious cold-chain foods will be dialectically discussed.

### 4.1. Nucleic Acid Test

SARS-CoV-2 is a positive-sense RNA virus, and the feature gene can be used as target analytes [49]. Nucleic acid detection technology is the most direct and essential pathogenic evidence for food contamination (Figure 2A). It has the advantages of early diagnosis, high sensitivity, and good specificity, and is the gold standard for SARS-CoV-2 detection (e.g., RT-PCR) [50].

#### 4.1.1. PCR-Based Techniques

Among the nucleic acid detection methods, RT-PCR can effectively amplify trace viral genes in nucleic acid mixtures and has the characteristics of fast detection speed, high sensitivity, and strong specificity [82]. RT-PCR uses sequence-specific primers to identify tiny RNA targets. The recognized RNA is then transcribed by reverse transcriptase to cDNA, which is then used as a template for DNA replication through PCR. However, the measurement is an enzyme-dependent multi-step technique, and the operation is complicated. The turnaround time takes a few hours which cannot meet the requirements of rapid testing of cold-chain food [83]. The test facilities and instruments are not portable, and the collected samples need to be transported to the laboratory for testing. The samples may produce false-negative results due to improper collection or processing. Improper operation or insufficient laboratory conditions may cause false positives due to aerosol contamination [84,85,86]. One-step nested RT-PCR is a flexible and easy method to test SARS-CoV-2. If the coronavirus mutates in one key amplified nucleotide, at least one pair can still be amplified [87]. The detection cost is lower than RT-PCR, but nested PCR is not feasible as a detection method for cold-chain food, because it is time-consuming and has a high risk of cross-contamination. In contrast, repetitive digital PCR is less interfered with by background wild DNA molecules, so it can reduce the impact of non-target DNA during cold-chain detection [88]. Digital PCR has obvious advantages when the viral load of cold-chain food samples is low or the sample nucleic acid is degraded. However, the cost of digital PCR detection is relatively high and the instrument is not portable [89]. It also involves the usage and storage of enzyme reagents, which will increase the cost and technological hurdles of detection. The digital PCR operation process has the disadvantage of being easily contaminated. To avoid false-positive results, it is necessary to establish strict internal quality control specifications for the laboratory and strictly regulate the testing operation process [90,91].

#### 4.1.2. RT-LAMP

Reverse transcription loop-mediated isothermal amplification (RT-LAMP) is a nucleic acid amplification assay, which is characterized by multiple specific primers for the target gene at a constant temperature of 60–65 °C under the DNA polymerase. About 10^9^-10^10^ times of nucleic acid amplification can be achieved in 15–60 min. RT-LAMP has the characteristics of simple operation, strong specificity (2 to 5 orders of magnitude higher than traditional PCR methods), and easy product detection [92]. The RT-LAMP results can be judged by visually observing the generation of white turbidity or green fluorescence. It is simple, quick, and does not require gel electrophoresis like PCR. As a point-of-care testing (POCT)-type nucleic acid detection method, LAMP requires less professional equipment, such as a thermal cycler, and the price of the instrument is lower than qRT-PCR, which can well meet the real-time requirements of SARS-CoV-2 detection on cold-chain food [93]. The detection sensitivity of RT-LAMP can reach 10 copies, and it has high specificity [94]. RT-LAMP is highly suitable for detecting > 60 copies/10 μL in sample. However, the test involves the use of enzymes (e.g., recombinase) that the storage of enzyme reagent needs, resulting in extra cost. When the test is performed for a long time, non-specific amplification may produce false-positive results if cold-chain food sampling < 10 copies/10 μL [95]. Based on RT-LAMP, a portable and scalable laser-engraved microwell array chip for multiplex amplification of viral RNA samples has been developed, which is a promising device for SARS-CoV-2 detection on cold-chain food.

#### 4.1.3. CRISPR-Based System

CRISPR-Cas is a nonspecific RNA system that can be activated by the amplified product RNA, cleavages the reporter RNA, and releases a fluorescent dye from the quencher. The CRISPR-based system has attracted growing enthusiasm due to its pathogen diagnosis ability [96]. It can realize the on-site test of SARS-CoV-2 on cold-chain food using simple equipment. The test time varies from 40 to 70 min when excluding the time for RNA extraction [97,98]. Combined with RT-LAMP technology, the CRISPR-Cas system can achieve an LoD of 10 copies/μL. Most CRISPR-based SARS-CoV-2 detection methods use the Cas12 enzyme to specifically recognize the virus sequence [99]. In addition, the all-in-one dual CRISPR-Cas12a analysis system does not need a pre-amplification step and it improves the sensitivity of the assay by using double CRISPR RNA. It can detect 1.2 DNA targets and 4.6 RNA targets in 40 min. The system can be developed as a one-step test platform without the need for cDNA preparation which has the potential for SARS-CoV-2 detection on cold-chain food [100].

#### 4.1.4. Microfluidic Biochip

Microfluidic chips integrate various small-scale laboratory functions on a single chip to complete the steps in traditional laboratories [101]. It uses a small number of reagents and samples to obtain accurate test results in a short time, and is especially suitable for the rapid detection of SARS-CoV-2 on cold-chain food. Recently, paper-based microfluidics, centrifugal chips, wearable microfluidic devices, and digital nucleic acid detection chips have been proposed for pathogen testing and disease screening [102]. For instance, the IDNOW^®^ instrument proposed by Abbott™ in the United States can detect a positive sample. The product has received an emergency use authorization (EUA) from the U.S. Food and Drug Administration (FDA). The instrument weighs only 3 kg and is portable and suitable for POCT [103,104].

#### 4.1.5. Whole-Genome Sequencing

Whole-genome sequencing (WGS) is an effective tool to comprehensive understand SARS-CoV-2. The assay belongs to high-throughput sequencing, or next-generation sequencing, which is a culture-free, unbiased, direct extraction of DNA or RNA from clinical samples [52,53]. However, the operation steps of WGS are relatively complex and the operation technology requirements are high. The RNA can be extracted using the kit and whole-genome sequencing performed on an instrument (e.g., Illumina iSeq 100). Peculiarly, metagenomics is a high-sensitivity pan-pathogen assay for the discovery of novel pathogens and infectious disease diagnosis, which is applied in the simultaneous and rapid detection of SARS-CoV-2 [105]. Because the whole gene sequencing requires a professional operator, complex sample pretreatment, and long-term periods, other methods are usually combined to generate test reports.

### 4.2. Immunological Methods

SARS-CoV-2 has a wide mammalian host range, including minks, cows, white-tailed deer, dogs, domestic cats, swine, lions, etc. [106,107,108,109]. Some of these animals may serve as viral carriers once they are made into food-related products [110]. Immunological tests can directly detect the antigen biomarker of SARS-CoV-2 and can be used to determine whether food (e.g., animal products) is infected or contaminated by the virus. The immunological methods that mainly include antigen tests, serology tests, and cytokine storm diagnoses can be used for SARS-CoV-2 detection for these animal foods. For plant foods, that cannot undergo an immune response to produce antibodies, serum antibody immunological testing and cytokine storm diagnosis cannot be applied to the detection of cold-chain food unless the plant food is contaminated with the body fluids of an infected person [111]. In the following sections, we will describe each of the above methods in detail.

#### 4.2.1. Antigen Immunological Test

Antigen tests are the main immunological method for food quarantine. The structural proteins, such as the spike glycoprotein (S), envelope protein (E), membrane protein (M), and nucleocapsid protein (N) of SARS-CoV-2 are the primary antigens used for the immunological test. Among the antigen immunological test methods (Figure 2B), ELISA is a highly sensitive immunological experimental technique that combines an antigen, antibody-specific reaction, and high-efficiency enzyme catalysis on the substrate. This detection method has high sensitivity and low difficulty in carrier standardization, but the detection steps are more cumbersome and easier to contaminate [112]. ELISA kits can test multiple samples in a single run; however, they lack point-of-care applicability and the non-specific binding of antibodies or antigens to the plate may lead to false positive results [113]. In addition to ELISA, the detection of trace S-protein (S1 subunit) for real-time SARS-CoV-2 detection is currently a known method that can be well applied to cold-chain foods. The S-protein particles can be attracted to the surface of the sensor and captured by the antibody within 20 s, which meets the real-time detection requirements on-site. The linear range is wide and covers the possible range of the concentration of S-protein on the food surface. The developed ultra-low LOD strategy has shown great advantages in the detection of virus markers with low concentrations of cold-chain foods [25]. The single sensor device can act as a disposable chip and its cost is estimated to be 1 US dollar. The operations of the device are relatively simple and can be operated by non-technical personnel [114]. Thus, the S-protein detection platform meets the requirements of rapid response, lower detection limit, high specificity, friendly operation, and low cost, which provides a promising solution for SARS-CoV-2 detection on cold-chain food [115].

Besides, the biosensor equipped with a chemical or biological receptor (e.g., antibody) can specifically interact with the target analyte showing a quantitative signal of the recognition process [116]. Compared with traditional laboratory methods, biosensors can provide a cheap, sensitive, rapid, miniaturized, and portable platform for SARS-CoV-2 detection which is promising for the on-site detection of cold-chain food contamination. Several studies have proved biosensor technology is convenient in SARS-CoV-2 S protein detection based on a bioelectrical identification assay [117]. The biosensor can detect S protein within 3 min with a LOD of about 1 fg/mL, and no cross-reaction with SARS-CoV-2 nucleocapsid protein was found. The portable readout system of the ready-made biosensor platform can be controlled by a smartphone or a tablet computer [118]. The high sensitivity, rapidness, and simplicity of biosensors make it a great advantage in the detection of viruses on cold-chain food. It can be easily controlled and has strong practical applicability to test contaminations on cold-chain food in factories or customs. However, some biosensors involve the use of enzyme reagents, and the biosensor technology is still in the development stage [119].

#### 4.2.2. Serum Antibody Immunological Test

Once the body is infected, the living organisms will produce specific antibodies, such as anti-SARS-CoV-2 IgM and IgG. Though animal food is frozen or transported in the cold-chain, the antibodies from an earlier infection can be preserved. The safety risk of infected COVID-19 is relatively low by eating cold-chain animal food and has not been studied to date. However, these serum antibodies can be used as the immunological target, especially in cold-chain animal food with a low viral load. Compared with nucleic acid testing, blood samples for antibody serology testing are easier to obtain, which greatly reduces the risk of infection of medical staff during specimen collection and testing, and makes it easier for primary laboratories to carry out screening work. For instance (Figure 2C), luciferase immunosorbent assay (LISA) is an easily and rapidly developed semi-quantitative method and is appropriate for detecting specific antibodies from cold-chain animal food [78]. Compared with enzyme-linked immunosorbent assays, DNA-assisted nanopore sensing assay can reliably quantify SARS-CoV-2 antibodies with high accuracy, wide dynamic range, and the potential for automated detection on cold-chain food [76]. Though modified with probe DNA to label IgG or IgM antibodies, the nanopore sensor can quantify the probe DNAs when thermal dehybridization of gold nanoparticles (AuNPs) probe DNAs was performed. In addition, surface plasmon resonance (SPR) biosensors assay adopts the optical detection method. Indirect aggregation can be used for virus detection by modifying targeted molecules on the virus surface. If you do not consider the portability of the SPR device, it may be the most promising technique for cold-chain food quarantine [75].

#### 4.2.3. Cytokine Storm Assay

No research so far has proposed cytokine storm detection on cold-chain foods or food packaging. However, living organisms infected with SARS-CoV-2 may experience high levels of release of inflammatory cytokines, such as interleukin-2 (IL-2), interleukin-6 (IL-6), interleukin-7 (IL-7), interleukin-10 (IL-10), granulocyte colony-stimulating factor (G-CSF), interferon-inducible protein-10 (IP-10), macrophage chemoattractant protein-1 (MCP-1), macrophage inflammatory protein 1α (MIP-1α), and tumor necrosis factor-alpha (TNFα) [120]. Thus, the immunogenic cytokine storm can act as the detective target of SARS-CoV-2 in cold-chain animal food (Figure 2D). For instance, the SARS-CoV-2 antigens were used to determine the functional T-cell responses by using an enzyme-linked immunosorbent spot (ELISpot) interferon-γ release assay. The assay has high diagnostic sensitivity to determine the protective immunity after COVID-19 [121]. Using Luminex technology, the expression levels of cytokines IL-2, interferon γ (IFN-γ), TNF-α, IL-4, IL-6, IL-10, and interleukin-17A (IL-17A) in serum were detected by the technique to reflect the disease condition. This method is comparable to ELISA in precision but has a wider detection range, higher sensitivity, and much lower specimen dosage than ELISA [122].

## 5. Discussion and Conclusions

At present, countries have adopted different epidemic prevention policies, some through closed management of patients to prevent more people from getting sick, while actively vaccinating; others plan to coexist with SARS-CoV-2. This imbalance will be challenging for countries that persist in lockdown policies. Of course, human-to-human transmission is still the main mode of transmission, but the characteristics of food cold-chain transmission are already beginning to show. Taking active measures to prevent the spread of COVID-19 over long distances will be a necessity. Therefore, this paper predicts the development trend of the COVID-19 pandemic based on probability and critically analyzes the existing detection methods.

Firstly, SARS-CoV-2 is highly stable on fish, meat, poultry, pigskin, and other foods under cold storage (4 °C). It can stay on the packaging of cold-chain products or their surface for a longer time, and transportation of these products can lead to viral spreading. This caused a huge risk of long-distance transmission through contaminated cold-chain foods. Secondly, China has reported a few infections, especially asymptomatic infections, caused by imported cold-chain products in Tianjin, Qingdao, Dalian, Beijing, etc. Live coronavirus has been detected and isolated in the package of imported frozen cod. Notably, the virus identified in the stevedores is highly homologous to the disease virus on the packaging of frozen cod. COVID-19 is likely to have spread from the environment to humans via cold-chain logistics from contaminated imported food. These findings signal that the virus can be carried over long distances across borders with contaminated cold-chain foods. Despite the partial uncertainties for cold-chain transmission, SARS-CoV-2 can survive in the cold-chain, and eliminating these contaminated cold-chain foods through effective quarantine measures may therefore be sensible. More research on the frequency of SARS-CoV-2 contamination on food packaging, the association between detection and infectious virus, and SARS-CoV-2 viability and infectivity in conditions that simulate those found in cold-chain logistics is warranted.

In summary, although no evidence suggests that SARS-CoV-2 causes foodborne illness, the COVID-19 pandemic causes a severe threat to food safety and public health. In the post-pandemic era, the imported cases in the supply chain still have the opportunity to cause the local outbreaks of the COVID-19 epidemic. Considerable attention should be paid to the cross-border spread of COVID-19 caused by cold-chain foods. Monitoring experience would be useful in terms of preventing the emergence of various foodborne infections. Systematic research that focuses on the formation, transmission, and solution of cold-chain food contamination is still rare. Every link in the farm-to-table process is worthy of attention. The disinfection of raw materials and packaging is necessary for this period of fighting against COVID-19. Food safety education and strict hygiene conditions are required for the staff involved in the process. For foods from countries with looser controls for cold-chain foods (poultry, seafood, cubed meats, etc.), the most critical aspect of import control is virus detection at customs. At present, the mainstream food detection method is RT-PCR, but it cannot meet the requirements of real-time, sensitive, specific, portable, on-site detection, and large-scale application of food virus detection. Although the impact of food contact on the SARS-CoV-2 transmission is relatively low and many test methods will likely never be implemented in practice, there is still a great necessity in the real supply chain to stop the spread of the virus through efficient screening measures.

## Figures and Tables

**Figure 1 foods-11-01540-f001:**
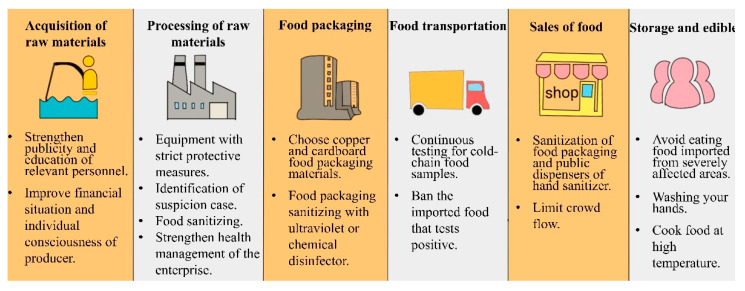
The possible solutions for the potential SARS-CoV-2 contaminated risks during the six links in the cold-chain.

**Figure 2 foods-11-01540-f002:**
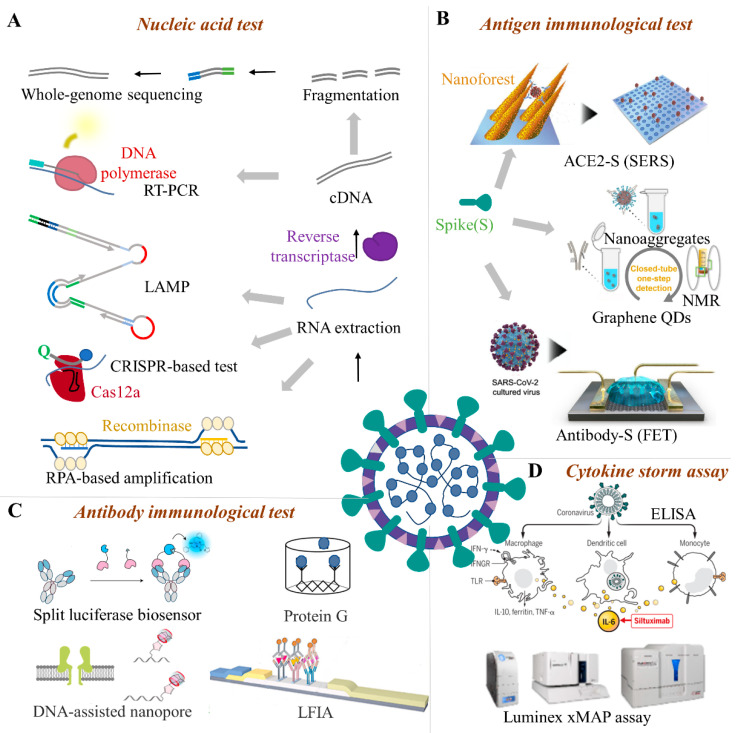
The advanced quarantine methods for SARS-CoV-2. For SARS-CoV-2 quarantine, many test techniques including nucleic acid and immunological methods are available. Nucleic acid tests include whole-genome sequencing and specific gene detection (**A**). Immunological tests include antigen tests (**B**), antibody immunological tests (**C**), and cytokine storm diagnoses (**D**). RT-PCR(Reverse transcription-polymerase chain reaction), LAMP (Loop-mediated isothermal amplification), CRISPR (Clustered Regularly Interspaced Short Palindromic Repeats), LFIA (Lateral-flow immunochromatographic assay), FET (Field-effect transistor), ELISA (Enzyme linked immunosorbent assay), xMAP (Multi-analyte profiling), SERS(Surface-enhanced raman scattering).

**Table 1 foods-11-01540-t001:** The advance of SARS-CoV-2 test techniques.

Methods	Category	Subcategory	LOD	Specificity	Sensitivity	Cost	Time	Description	References
Nucleic acid test	RT-PCR		1–10 copies	97.06–99.69%	91.06–99.96%	USD 25–200	4-6 h	The gold standard for SARS-CoV-2 diagnosis is suitable for the large-scale test but needs specialized laboratory equipment and trained technicians	[51,52]
	Whole-genome sequencing		ND	ND	98.33–99.83%	USD 2000	48-72 h	The first complete genomic sequences of SARS-CoV-2 were obtained through metatranscriptomics approaches	[53,54]
	Isothermal amplification technology	Transcriptional colorimetric loop-mediated isothermal amplification	100 copies/μL	100%	85%	ND	21 h	Effectively reduce the false positive rate and improve the detection efficiency	[55]
		Proofreading enzyme-mediated isothermal amplification	100 copies	Effectively distinguish SARS-CoV-2 from SARS-CoV	Effectively detect as few as 100 copies of gene N RNA in 1 h	ND	50 min	Show similar analytical performance with the conventional RT-PCR	[56]
		Emulsion loop-mediated isothermal amplification	10, 10^3^, and 10^5^ copies/μL	ND	ND	ND	5–10 min	Limit of detection of 1 copy per microliter sample and portable device using a miniature spectrometer or a smartphone	[57]
	Recombinase polymerase amplification (RPA)	Combined RPA with rkDNA-graphene oxide probing system	6.0 aM	ND	ND	ND	1.6 h	Exhibit high selectivity and sensitivity for the diagnosis of COVID-19	[58]
		Recombinase polymerase amplification	7.659 copies/μL	100%	98%	USD 4.3	5–20 min	High specificity	[59,60]
		Isothermal RPA-lateral flow detection	0.25–2.5 copies/μL	100%	94%	ND	5 min	The detection limit of RPA-LF for SARS-CoV-2 was 35.4 nucleocapsid (N) gene copies/L; the sensitivity was similar to that of qualitative real-time PCR	[61]
	Hybrid capture immunofluorescence assay	Hybrid capture immunofluorescence assay	500 copies per mL	99%	100%	ND	45 min	The detection sensitivity is consistent with similar products on the market; however, this technique can only give qualitative results	[62]
		Entropy-driven amplified electrochemiluminescence	2.67 fM	ND	ND	ND	10–20 h	High selectivity and stability	[63]
	CRISPR-based test	Cas12a	10 copies per μL reaction	100%	95%	USD 6	40–60 min	Enables rapid, ultrasensitive (few copies), and highly specific nucleic acid detections	[43]
		Cas13a	10–100 copies per μL	100%	96%	USD 3.5	40–57 min	Rapid, sensitive, and with low instrument requirement	[64]
		Pyrococcus furiosus Argonaute coupled with modified ligase chain reaction	10 aM	ND	ND	Cheaper than CRISPR	~70 min	High sensitivity, high specificity, and multiplexing detection; without the use of RNA as guidance	[65]
Immunological test	Antigen immunological test	Quantum dot immunochromatographic assay	4.9 pg/mL	100%	75 pg/mL	USD 1.5	3 min	One single test that can cover hs-CRP and routine-range CRP with a detection range from 1 to 200 μg mL^−1^	[66]
		QuickNavi™-COVID-19 Ag immunochromatographic test	ND	100%	86.7%	Cheaper than nucleic acid amplification tests	5 min	The overall sensitivity was 86.7%, and the positive detection rate in patients with CT < 30 was comparable to that of RT-PCR	[67]
		Magnetic graphene quantum dots	248 Particles mL^−1^	Related to SARS-CoV-2 antigen protein	No response to MERS-CoV	USD 1.25	2 min	Sensitive detection without sample pretreatment in one step with a LOD of 248 Particles mL^−1^	[68]
		Binax-CoV2	1.6 × 10^4^–4.3 × 10^4^ viral RNA copies	99.9%	93.3%	USD 5	15 min	The sensitivity of Binax-CoV2 was 93.3% and the specificity was 99.9%	[69]
		SERS biosensor	80 copies mL^−1^	Related to the sensing environment	Suffers from non-specific binding	More expensive than ELISA	5 min	The low detection limit (LOD) can be reduced to 80 parts mL^−1^	[70]
		Interdigitated microelectrode chip	2.29 × 10^−6^ ng/mL	4.27 × 10^−4^ ng/mL	234:1	USD 1	20 s	The linear range is 10^−5^–10^−1^ ng/mL; the strategy is real-time, sensitive, selective, and large-scale in cold-chain food quarantine	[25]
	Serum antibody immunological test	Split luciferase antibody biosensors	ND	> 99%	> 98%	∼15 ¢	5 min	The sensitivity to detect anti-S protein antibodies was 89% and anti-N protein antibodies were 98%, and the specificity of both was more than 99%	[71]
		Colloidal gold immunochromatography assay	20.00 IU/mL	96.2%	71.1%	ND	10–15 min	The IgM/IgG test assay demonstrated high sensitivity of 71.1% and specificity of 96.2% in 150 suspect COVID-19 cases	[72]
		Chemiluminescence immunoassay	0.5–1.5 AU/mL	97.5%	78.65%	ND	1 h	The antibody detection rate has high sensitivity, high precision, quantitative detection, and easy automation	[73]
		Upconverting phosphor immunochromatography assay	ND	99.75%	89.15%	ND	10 min	High sensitivity, no interference from the background, and good stability	[74]
		Surface plasmon resonance biosensors (SPRS)	0.22 pM	ND	ND	ND	ND	The SPR biosensor is feasible in the concentration range of 2 to 1000 ng/mL	[75]
		DNA-assisted nanopore sensing	50 ng/mL (IgM) 10 ng/mL(IgG)	ND	ND	USD 8	ND	High sensitivity and specificity compared to laboratory techniques	[76]
		Colorimetric-fluorescent dual-mode lateral flow immunoassay biosensor	10 ng/mL(IgM) 5 ng/mL(IgG)	100%	ND	ND	ND	The combined detection sensitivity and specificity of this assay for IgM/IgG is 100%, and it has great potential for rapid and accurate detection	[76]
		The lateral flow immunoassay method	ND	90.63%	88.66%	ND	15 min	The limits of detection for IgM and IgG were 10 ng/mL and 5 ng/mL, respectively	[77]
		Luciferase immunosorbent assay (LISA)	0.4–75 pg / μl	100%	71%	ND	~60 min	LISA had a sensitivity of 71% in COVID-19 patients and a specificity of 100% in healthy blood donors in the second week after onset	[78]
		Enzyme-linked immunosorbent assays	0.095 (IgM) 0.083 (IgG)	ND	98%	ND	80–120 min	High sensitivity and specificity	[79]
		Enzyme-linked immunosorbent assays	ND	88.2–99.2% (IgM) 75.6–98.3% (IgG)	78.2% (IgG) 96.6%(IgM)	ND	1.5 h	ELISA was used to detect IgG antibodies in confirmed patients with COVID-19, and the sensitivity to detect IgM antibodies was low	[80]
		Enzyme-linked immunosorbent assays (ELISA)	ND	93–100%	65–85%	ND	4 h	The detection precision is similar to ELISA, but the detection range is wider and the sensitivity is higher	[81]

Note that ND is not defined in the literature. LOD: limit of detection. RT-PCR(Reverse transcription-polymerase chain reaction). CRISPR(Clustered Regularly Interspaced Short Palindromic Repeats), SERS(Surface-enhanced raman scattering). aM: 10^−18^ mol/mL. fM: 10^−15^ mol/mL. IU: international unit. AU: arbitrary unit.

## Data Availability

The data presented in this study are available on request from the corresponding author.

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
