# Peer review of "The Safety of Cold-Chain Food in Post-COVID-19 Pandemic: Precaution and Quarantine"

_foods, 2022, doi:10.3390/foods11111540_

Round 1

Reviewer 1 Report

The authors have answered to my comments. 

Author Response

Thanks for the reviewers’ helpful comments. We have made revisions according to the reviewers’ comments and suggestions, and the amendments are highlighted in yellow in the revised manuscript. Point-to-point response to the reviewers’ comments are attached as below.

The authors have answered to my comments.

Answer: we would like to express our sincere thanks for your positive comments.

Reviewer 2 Report

Even though the paper title is rather general, the paper is mostly concerned with the testing issue. Please adjust the title accordingly.

The table is unclear in the sense of the test costs, some labels are 'not high' etc. which does not provide meaningful information and even prevents from a comparison. Therefore, a certain scale could be used to make the rows comparable (in case exact cost is not known). The capitalization should also be checked. Some cells for LOD are empty.

Is pooled testing possible? Then, the costs may be reduced.

Are the test reported in the table related to food only? Connection to the food problematique remains unclear.

Author Response

Thanks for the reviewers’ helpful comments. We have made revisions according to the reviewers’ comments and suggestions, and the amendments are highlighted in yellow in the revised manuscript. Point-to-point response to the reviewers’ comments are attached as below.

  1. Even though the paper title is rather general, the paper is mostly concerned with the testing issue. Please adjust the title accordingly.

Answer: We have adjusted the title to “The Safety of Cold-chain Food in post-COVID-19 Pandemic: Inspection and Quarantine” according to your suggestion.

  1. The table is unclear in the sense of the test costs, some labels are 'not high' etc. which does not provide meaningful information and even prevents from a comparison. Therefore, a certain scale could be used to make the rows comparable (in case exact cost is not known). The capitalization should also be checked. Some cells for LOD are empty.

Answer: We have supplemented the LOD, specificity, sensitivity, test costs, and the time of different SARS-CoV-2 test techniques based on the referred literature. We use the symbol ND to represent the item not defined in the literature. The updated table is below.

Table 1 The advance of SARS-CoV-2 test techniques.

Methods

Category

Subcategory

LOD

Specificity

Sensitivity

Cost

Time

Description

References

Nucleic acid test

RT-PCR

1–10 copies

97.06-99.69%

91.06-99.96%

$ 25-200

4~6 h

The gold standard for SARS-CoV-2 diagnosis is suitable for the large-scale test but needs specialized laboratory equipment and trained technicians

[51,52]

Whole-genome sequencing

ND

ND

98.33%-99.83%

$ 2000

48~72 h

The first complete genomic sequences of SARS-CoV-2 were obtained through metatranscriptomics approaches

[53,54]

Isothermal amplification technology

Recombinase polymerase amplification (RPA)

Transcriptional Colorimetric Loop-Mediated Isothermal Amplification

100 copies/μL

100%

85%

ND

21 h

Effectively reduce the false positive rate and improve the detection efficiency

[55]

Proofreading enzyme-mediated isothermal amplification

100 copies

Effectively distinguish SARS-COV-2 from SARS-COV

Effectively detect as few as 100 copies of gene N RNA in 1 hour

ND

50 min

Show similar analytical performance with the conventional RT-PCR

[56]

Emulsion loop-mediated isothermal amplification

10, 103, and 105 copies/μL

ND

ND

ND

5-10 min

Limit of detection of 1 copy per microliter sample and portable device using a miniature spectrometer or a smartphone

[57]

Combined RPA with rkDNA-graphene oxide probing system

6.0 aM

ND

ND

ND

1.6 h

Exhibit high selectivity and sensitivity for the diagnosis of COVID-19

[58]

Recombinase Polymerase Amplification

7.659 copies/μL

100%

98%

$ 4.3

5–20 min

High specificity

[59,60]

Isothermal RPA-lateral Flow Detection

0.25-2.5 copies/μL

100%

0-94%

ND

5 min

The detection limit of RPA-LF for SARS-CoV-2 was 35.4 nucleocapsid (N) gene copies / L. The sensitivity was similar to that of qualitative real-time PCR.

[61]

Hybrid capture immunofluorescence assay

Hybrid capture immunofluorescence assay

500 copies per ml 

99%

100%

ND

45 min

The detection sensitivity is consistent with similar products on the market. However, this technique can only give qualitative results

[62]

Entropy-driven amplified electrochemiluminescence

2.67 fM

ND

ND

ND

10 - 20 h

High selectivity and stability

[63]

CRISPR-based test

Cas12a

10 copies per μl reaction

100%

95%

$ 6

40-60 min

Enables rapid, ultrasensitive (few copies), and highly specific nucleic acid detections

[43]

Cas13a

10–100 copies per μL

100%

96%

$ 3.5

40-57 min

Rapid, sensitive, and with low instrument requirement

[64]

Pyrococcus furiosus Argonaute coupled with modified Ligase Chain Reaction

10 aM

ND

ND

Cheaper than CRISPR

~ 70 min

High sensitivity, high specificity, and multiplexing detection; Without the use of RNA as guidance

[65]

Immunological test

Antigen immunological test

Quantum Dot Immunochromatographic Assay

4.9 pg/mL

100%

75 pg

/mL

$ 1.5

3 min

One single test that can cover hs-CRP and routine-range CRP with a detection range from 1 to 200 μg mL−1

[66]

QuickNavi™-COVID19 Ag immunochromatographic test

ND

100%

86.7%

Cheaper than nucleic acid amplification tests

5 min

The overall sensitivity was 86.7%, and the positive detection rate in patients with CT < 30 was comparable to that of RT-PCR

[67]

Magnetic graphene quantum dots

248 Particles mL‒1

Related to SARS-CoV-2 antigen protein

No response to MERS-CoV

$1.25

2 min

Sensitive detection without sample pretreatment in one step with a LOD of 248 Particles mL‒1

[68]

Binax-CoV2

1.6×104-4.3×104 viral RNA copies

99.9%

93.3%

$ 5

15 min

The sensitivity of Binax-CoV2 was 93.3% and the specificity was 99.9%

[69]

SERS biosensor

80 copies mL-1

Related to the sensing environment

Suffers from non-specific binding

More expensive than ELISA

5 min

The low detection limit (LOD) can be reduced to 80 parts mL-1

[70]

Interdigitated microelectrode chip

2.29 × 10−6 ng/mL

4.27 × 10−4 ng/mL

234:1

$ 1

20 s

The linear range is 10−5–10−1 ng/mL. The strategy is real-time, sensitive, selective, and large-scale in cold-chain food quarantine.

[25]

Serum antibody immunological test

Split luciferase antibody biosensors

ND

>99%

>98%

∼15 ¢

5 min

The sensitivity to detect anti-S protein antibodies was 89% and anti-N protein antibodies were 98%, and the specificity of both was more than 99%

[71]

Colloidal gold immunochromatography assay

20.00 IU/ml

96.2%

71.1%

ND

10-15 min

The IgM/IgG test assay demonstrated high sensitivity of 71.1% and specificity of 96.2% in 150 suspect COVID-19 cases

[72]

Chemiluminescence immunoassay

0.5-1.5 AU/mL

97.5%

78.65%

ND

1 h

The antibody detection rate has high sensitivity, high precision, quantitative detection, and easy automation

[73]

Upconverting phosphor immunochromatography assay

ND

99.75%

89.15%

ND

10 min

High sensitivity, no interference from the background, and good stability

[74]

Surface plasmon resonance biosensors(SPRS)

0.22 pM

ND

ND

ND

ND

The SPR biosensor is feasible in the concentration range of 2 to 1000 ng/mL

[75]

DNA-assisted nanopore sensing

50 ng/mL (IgM) 10 ng/ml(IgG)

ND

ND

$ 8

ND

High sensitivity and specificity compared to laboratory techniques

[76]

Colorimetric-fluorescent dual-mode lateral flow immunoassay biosensor

10 ng/mL(IgM) 5 ng/ml(IgG)

100%

ND

ND

ND

The combined detection sensitivity and specificity of this assay for IgM/IgG is 100%, and it has great potential for rapid and accurate detection

[76]

The lateral flow immunoassay method

ND

90.63%

88.66%

ND

15 min

The limits of detection for IgM and IgG were 10 ng/mL and 5 ng/mL, respectively.

[77]

Luciferase immunosorbent assay (LISA)

0.4 - 75 pg / μl

100%

71%

ND

~60 min

LISA had a sensitivity of 71% in COVID-19 patients and a specificity of 100% in healthy blood donors in the second week after onset

[78]

Enzyme-linked immunosorbent assays

0.095 (IgM) 0.083 (IgG)

ND

98%

ND

80-120 min

High sensitivity and specificity

[79]

Enzyme-linked immunosorbent assays

ND

88.2%-99.2% (IgM) 75.6%-98.3% (IgG)

78.2% (IgG) 96.6%(IgM)

ND

1.5 h

ELISA was used to detect IgG antibodies in confirmed patients with COVID-19, and the sensitivity to detect IgM antibodies was low

[80]

Enzyme-linked immunosorbent assays (ELISA)

ND

93-100%

65%-85%

ND

4 h

The detection precision is similar to ELISA, but the detection range is wider and the sensitivity is higher

[81]

Note that ND is not defined in the literature. LOD: limit of detection. aM: 10-18 mol/mL. fM: 10-15 mol/ mL. IU: International Unit. AU: Arbitrary Unit.

References

  1. Böger, M.M. Fachi, R.O. Vilhena, A.F. Cobre, F.S. Tonin, R. Pontarolo. Systematic review with meta-analysis of the accuracy of diagnostic tests for COVID-19. AM J INFECT CONTROL.2021, 49, 21-29. DOI:https://doi.org/10.1016/j.ajic.2020.07.011.
  2. Chaimayo, B. Kaewnaphan, N. Tanlieng, N. Athipanyasilp, R. Sirijatuphat, M. Chayakulkeeree, N. Angkasekwinai, R. Sutthent, N. Puangpunngam, T. Tharmviboonsri. Rapid SARS-CoV-2 antigen detection assay in comparison with real-time RT-PCR assay for laboratory diagnosis of COVID-19 in Thailand. Virol. J. 2022,17 ,1-7. DOI:10.1186/s12985-020-01452-5.
  3. Greninger, A.L.; Zerr, D.M.; Qin, X.; Adler, A.L.; Sampoleo, R.; Kuypers, J.M.; Englund, J.A.; Jerome, K.R. Rapid Metagenomic Next-Generation Sequencing during an Investigation of Hospital-Acquired Human Parainfluenza Virus 3 Infections. J. Clin. Microbiol. 2017, 55, 177-182, DOI:10.1128/jcm.01881-16.
  4. Chiara, M.; D'Erchia, A.M.; Gissi, C.; Manzari, C.; Parisi, A.; Resta, N.; Zambelli, F.; Picardi, E.; Pavesi, G.; Horner, D.S.; et al. Next generation sequencing of SARS-CoV-2 genomes: challenges, applications and opportunities. Brief. Bioinform. 2021, 22, 616-630, DOI: 1093/bib/bbaa297.
  5. Ji, C.; Xue, S.; Yu, M.; Liu, J.; Zhang, Q.; Zuo, F.; Zheng, Q.; Zhao, L.; Zhang, H.; Cao, J. Rapid Detection of SARS-CoV-2 Virus Using Dual Reverse Transcriptional Colorimetric Loop-Mediated Isothermal Amplification. ACS omega. 2021, 6, 8837-8849. DOI: https://doi.org/10.1021/acsomega.0c05781.
  6. Ding, S.; Chen, G.; Wei, Y.; Dong, J.; Du, F.; Cui, X.; Huang, X.; Tang, Z. Sequence-specific and multiplex detection of COVID-19 virus (SARS-CoV-2) using proofreading enzyme-mediated probe cleavage coupled with isothermal amplification. Biosens Bioelectron. 2021, 178, 113041, DOI: 10.1016/j.bios.2021.113041.
  7. Day, A.S.; Ulep, T.H.; Safavinia, B.; Hertenstein, T.; Budiman, E.; Dieckhaus, L.; Yoon, J.Y. Emulsion-based isothermal nucleic acid amplification for rapid SARS-CoV-2 detection via angle-dependent light scatter analysis. Biosens Bioelectron. 2021, 179, 113099, DOI: 1016/j.bios.2021.113099.
  8. Choi, M.H.; Lee, J.; Seo, Y.J. Combined recombinase polymerase amplification/rkDNA-graphene oxide probing system for detection of SARS-CoV-2. Anal. Chim. Acta. 2021, 1158, 338390, DOI: 1016/j.aca.2021.338390.
  9. Daher, R.K.; Stewart, G.; Boissinot, M.; Bergeron, M.G. Recombinase Polymerase Amplification for Diagnostic Applications. Clin. Chem. 2016, 62, 947-958, DOI: 1373/clinchem.2015.245829.
  10. Esbin, M.N.; Whitney, O.N.; Chong, S.; Maurer, A.; Darzacq, X.; Tjian, R. Overcoming the bottleneck to widespread testing: a rapid review of nucleic acid testing approaches for COVID-19 detection. RNA 2020, 26, 771-783, DOI:10.1261/rna.076232.120.
  11. Shelite, T.R.; Uscanga-Palomeque, A.C.; Castellanos, A.; Melby, P.C.; Travi, B.L. Isothermal recombinase polymerase amplification-lateral flow detection of SARS-CoV-2, the etiological agent of COVID-19. J. Virol. Methods. 2021. DOI: https://doi.org/10.1016/j.jviromet.2021.114227.
  12. Wang, D.; He, S.; Wang, X.; Yan, Y.; Liu, J.; Wu, S.; Liu, S.; Lei, Y.; Chen, M.; Li, L. Rapid lateral flow immunoassay for the fluorescence detection of SARS-CoV-2 RNA. Nat. Biomed. Eng. 2020, 1-9. DOI: https://doi.org/10.1038/s41551-020-00655-z.
  13. Fan, Z.; Yao, B.; Ding, Y.; Zhao, J.; Xie, M.; Zhang, K. Entropy-driven amplified electrochemiluminescence biosensor for RdRp gene of SARS-CoV-2 detection with self-assembled DNA tetrahedron scaffolds. Biosens. Bioelectron. 2021, 178, 113015, DOI: 1016/j.bios.2021.113015.
  14. Hou, T.; Zeng, W.; Yang, M.; Chen, W.; Ren, L.; Ai, J.; Wu, J.; Liao, Y.; Gou, X.; Li, Y. Development and evaluation of a rapid CRISPR-based diagnostic for COVID-19. Plos. Pathog. 2020, 16, e1008705. DOI: https://doi.org/10.1371/journal.ppat.1008705.
  15. Wang, L.; He, R.; Lv, B.; Yu, X.; Liu, Y.; Yang, J.; Li, W.; Wang, Y.; Zhang, H.; Yan, G.; et al. Pyrococcus furiosus Argonaute coupled with modified ligase chain reaction for detection of SARS-CoV-2 and HPV. Talanta 2021, 227, 122154. DOI: 1016/j.talanta.2021.122154.
  16. Zhang, J. Li, Y. Li, G. Tan, M. Sun, Y. Shan, Y. Zhang, X. Wang, K. Song, R. Shi, L. Huang, F. Liu, Y. Yi, X. Wu. SARS-CoV-2 detection using quantum dot fluorescence immunochromatography combined with isothermal amplification and CRISPR/Cas13a. Biosens. Bioelectron.2022, 202,113978. DOI: https://doi.org/10.1016/j.bios.2022.113978.
  17. Takeuchi, Y. Akashi, D. Kato, M. Kuwahara, S. Muramatsu, A. Ueda, S. Notake, K. Nakamura, H. Ishikawa, H. Suzuki. The evaluation of a newly developed antigen test (QuickNavi™-COVID19 Ag) for SARS-CoV-2: A prospective observational study in Japan.J INFECT CHEMOTHER. 2021,27,890-894. DOI: 10.1016/j.jiac.2021.02.029.
  18. Li, Y.; Ma, P.; Tao, Q.; Krause, H.J.; Yang, S.; Ding, G.; Dong, H.; Xie, X. Magnetic graphene quantum dots facilitate closed-tube one-step detection of SARS-CoV-2 with ultra-low field NMR relaxometry. Actuat. B-Chem. 2021, 337, 129786, DOI: 10.1016/j.snb.2021.129786.
  19. Seo, G. Lee, M.J. Kim, S.-H. Baek, M. Choi, K.B. Ku, C.-S. Lee, S. Jun, D. Park, H.G. Kim. Rapid detection of COVID-19 causative virus (SARS-CoV-2) in human nasopharyngeal swab specimens using field-effect transistor-based biosensor. ACS Nano. 2020,14 ,5135-5142. DOI: https://doi.org/10.1021/acsnano.0c02823.
  20. Yadav, M.A. Sadique, P. Ranjan, N. Kumar, A. Singhal, A.K. Srivastava, R. Khan. SERS based lateral flow immunoassay for point-of-care detection of SARS-CoV-2 in clinical samples. ACS Appl. Bio Mater. 2021, 4 ,2974-2995, DOI: https://doi.org/10.1021/acsabm.1c00102.
  21. Elledge, S.K.; Zhou, X.X.; Byrnes, J.R.; Martinko, A.J.; Lui, I.; Pance, K.; Lim, S.A.; Glasgow, J.E.; Glasgow, A.A.; Turcios, K.; et al. Engineering luminescent biosensors for point-of-care SARS-CoV-2 antibody detection. Biotechnol. 2020, DOI:10.1101/2020.08.17.20176925.
  22. Wang, Q. Du, B. Guo, D. Mu, X. Lu, Q. Ma, Y. Guo, L. Fang, B. Zhang, G. Zhang. A method to prevent SARS-CoV-2 IgM false positives in gold immunochromatography and enzyme-linked immunosorbent assays. J. CLIN MICROBIOL. 2020, 58 , e00375-00320. DOI: https://doi.org/10.1128/JCM.00375-20.
  23. Padoan, A.; Cosma, C.; Sciacovelli, L.; Faggian, D.; Plebani, M. Analytical performances of a chemiluminescence immunoassay for SARS-CoV-2 IgM/IgG and antibody kinetics. Clin. Chem. Lab. Med. 2020, 58, 1081-1088, DOI:10.1515/cclm-2020-0443.
  24. Niedbala, R.S.; Feindt, H.; Kardos, K.; Vail, T.; Burton, J.; Bielska, B.; Li, S.; Milunic, D.; Bourdelle, P.; Vallejo, R. Detection of analytes by immunoassay using up-converting phosphor technology. Biochem. 2001, 293, 22-30, DOI:10.1006/abio.2001.5105.
  25. Qiu, Z. Gai, Y. Tao, J. Schmitt, G.A. Kullak-Ublick, J. Wang. Dual-functional plasmonic photothermal biosensors for highly accurate severe acute respiratory syndrome coronavirus 2 detection. ACS nano.2020, 14 , 5268-5277.DOI: https://doi.org/10.1021/acsnano.0c02439
  26. Zhang, Z.; Wang, X.; Wei, X.; Zheng, S.W.; Lenhart, B.J.; Xu, P.; Li, J.; Pan, J.; Albrecht, H.; Liu, C. Multiplex quantitative detection of SARS-CoV-2 specific IgG and IgM antibodies based on DNA-assisted nanopore sensing. Biosens Bioelectron. 2021, 181, 113134, DOI:10.1016/j.bios.2021.113134.
  27. Bayin, Q.; Huang, L.; Ren, C.; Fu, Y.; Ma, X.; Guo, J. Anti-SARS-CoV-2 IgG and IgM detection with a GMR based LFIA system. Talanta 2021, 227, 122207, DOI:10.1016/j.talanta.2021.122207.
  28. Yao, L. Drecun, F. Aboualizadeh, S.J. Kim, Z. Li, H. Wood, E.J. Valcourt, K. Manguiat, S. Plenderleith, L. Yip. A homogeneous split-luciferase assay for rapid and sensitive detection of anti-SARS CoV-2 antibodies. Nat. Commun.2021, 12 , 1-8. DOI: https://doi.org/10.1038/s41467-021-22102-6.
  29. Bundschuh, C.; Egger, M.; Wiesinger, K.; Gabriel, C.; Clodi, M.; Mueller, T.; Dieplinger, B. Evaluation of the EDI enzyme linked immunosorbent assays for the detection of SARS-CoV-2 IgM and IgG antibodies in human plasma. Chim. Acta 2020, 509, 79-82, DOI:10.1016/j.cca.2020.05.047.
  30. Al-Jighefee H T, Yassine H M, Nasrallah G K. Evaluation of antibody response in symptomatic and asymptomatic COVID-19 patients and diagnostic assessment of new IgM/IgG ELISA kits[J]. Pathogens, 2021, 10(2): 161. DOI: https://doi.org/10.3390/pathogens10020161
  31. Skalnikova, H.K.; Kepkova, K.V.; Vodicka, P. Luminex xMAP Assay to Quantify Cytokines in Cancer Patient Serum. In Immune Mediators in Cancer; Springer, 2020, DOI: 10.1007/978-1-0716-0247-8_6.

  1. Is pooled testing possible? Then, the costs may be reduced.

Answer: Thanks for your suggestion. We have added pooled testing in the manuscript and the amendments are below. A pooled testing strategy for identifying SARS-CoV-2 can be used as it can handle a large number of food samples. Pooled testing benefits faster results, lower costs of testing, and fewer test kits which can greatly reduce test costs.

  1. Are the test reported in the table related to food only? Connection to the food problematique remains unclear.

Answer: The detection of trace S-protein described in chapter 4.2.1. Antigen immunological test is a currently known method that is only applied to cold chain foods. The S-protein particles can be attracted to the surface of the sensor and captured by the antibody within 20 seconds which meets the real-time detection requirements on site. The linear range is wide that covers the possible range of the concentration of S-protein on the food surface. The developed ultra-low LOD strategy has shown great advantages in the detection of virus markers with low concentrations of cold chain foods. (Ref. Zhang, J.; Fang, X.; Mao, Y.; Qi, H.; Wu, J.; Liu, X.; You, F.; Zhao, W.; Chen, Y.; Zheng, L. Real-time, selective, and low-cost detection of trace-level SARS-CoV-2 spike-protein for cold-chain food quarantine. NPJ Sci. Food 2021, 5, 12, DOI:10.1038/s41538-021-00094-3.). We have added these descriptions in Table 1 according to your suggestion.

Reviewer 3 Report

The article covers the important topic of food monitoring based on an example of Covid. This study contains many valuable and relevant points. 
The first part of the article consists of general information and contains safety recommendations for preventing the spread of the SARS-CoV-2 virus through the cold chain foods. This point is not widely discussed and it is truly valuable in the current state of affairs.
The second part presents an extended analysis of different SARS-CoV-2 testing technologies and an array of counter measures.
There are some recommendations to the authors from my point of view:
1. The conclusion lacks a solid statement. Authors should present more ways to solve the problem, rather than focusing on paths that led to this problem and the experience of how it was dealt with before. 
2. The value of the review can be increased. Authors should underline that while there is no conclusive evidence for foodborne spread of SARS-CoV-2 monitoring experience would be useful in terms of preventing the emergence of various foodborne infections. 
The article is relevant and reliable. The research is well written, exceptionally structural and understandable for a competent reader and even for a general audience. The analysis presented in the article may be useful for monitoring foodborne viral infections in the food supply chain.

Author Response

Thanks for the reviewers’ helpful comments. We have made revisions according to the reviewers’ comments and suggestions, and the amendments are highlighted in yellow in the revised manuscript. Point-to-point response to the reviewers’ comments are attached as below.

  1. The conclusion lacks a solid statement. Authors should present more ways to solve the problem, rather than focusing on paths that led to this problem and the experience of how it was dealt with before.

Answer: Thanks for your suggestion. We have added more ways to solve the problem in the conclusion part. Every link in the farm-to-table process is worthy of attention. The disinfection of raw materials and packaging is necessary for the period fighting against the COVID-19. Food safety education and strict hygiene conditions are required for the staff involved in the process.
2. The value of the review can be increased. Authors should underline that while there is no conclusive evidence for foodborne spread of SARS-CoV-2 monitoring experience would be useful in terms of preventing the emergence of various foodborne infections.

Answer: Thanks for your suggestion. We have added the above statement to the manuscript.
The article is relevant and reliable. The research is well written, exceptionally structural and understandable for a competent reader and even for a general audience. The analysis presented in the article may be useful for monitoring foodborne viral infections in the food supply chain.

Finally, we would like to express our sincere thanks for your constructive and positive comments. Thanks again!

This manuscript is a resubmission of an earlier submission. The following is a list of the peer review reports and author responses from that submission.

Round 1

Reviewer 1 Report

The paper discusses an important topic of COVID mitigation. The paper is quite elaborated, yet some improvements can be carried out.

The abstract should be more focused. The covid and sars-cov-2 are mentioned without clear linkage. The abstract mentions entry banks that are not directly related to the paper.

No figures are needed in introduction.

The paper does not describe the research strategy.

The title of Section 4 is unclear: why only characteristics of supply chain should be discussed? I think the major objective of the research is to discuss the measures for containing COVID in the supply chains.

The paper could refer to research on the economics and logistics underlying the supply chains, e.g. Baležentis, T., Morkūnas, M., Žičkienė, A., Volkov, A., Ribašauskienė, E., & Štreimikienė, D. (2021). Policies for Rapid Mitigation of the Crisis’ Effects on Agricultural Supply Chains: A Multi-Criteria Decision Support System with Monte Carlo Simulation. Sustainability13(21), 11899.

Reviewer 2 Report

This review is a timely work. Some major considerations:

(a) It would be good if the review is enhanced with the challenges/future perspectives with regards to the use of current technology for the detection of SARS-CoV-2 virus

(b) Comparisons of the use of current technology (maybe in the form of efficacy, reliability, validity and specificity) is always useful. 

(c) I am not able to find a single food items (poultry, seafoods, cubed meats etc) as the main topic of discussion. The whole manuscript was linearly-directed to the potential use of cold-chain quarantine technology. No background study/actual study have been included to summarise the use of current technology.

Reviewer 3 Report

In my opinion, the manuscript lacks sufficient scientific arguments to be published, since it puts on the same level facts that are proven with others that are absolutely in dispute, and that are even improbable.
The fact that for some time scientific journals have been especially receptive to publish any information related to COvid-19, has made the scientific literature on the subject increase in an illogical way that makes scientific information coexist with simple suppositions.

In fact, the authors themselves quote at the beginning of the conclusions "no evidence suggests that SRAS-Cov-2 causes foodborne illness". 

Consequently, the manuscript starts from a basic error that cannot be confirmed with the authors. 
Moreover, most of the article, together with the only table produced, deals with commercially available SARS-COv-2 detection methods, but not necessarily applied to food, packaging or any other activity related to food processing, distribution or consumption. the possible need for food quarantine, cited in the title (and therefore it is understood that it should be an important part of the manuscript), is only cited in a very superficial way in the final part of the manuscript.

Therefore, since the manuscript does not establish any conclusion, nor even a duly substantiated recommendation or hypothesis, it does not meet the characteristics to be published in a journal such as Foods.

This conclusion clashes head-on with the statement stated in lines 38 "cold-chain foods can act as the potential carrier of COVID-19", because this statement implies that SARS-COV-2 must be infective in order to spread COVID-19. Similar statements are scattered throughout the manuscript, as is the fact that Qingdao port workers have stated. It cannot be assumed that they have had no contact with COVID-19 because they are not aware of it. The same is true for some statements about risk from food procurement, packaging, food transport or consumption. It has not been demonstrated that any of these activities pose a real risk of contracting COVID-19.